# Sustainability Perception of Local Extra Virgin Olive Oil and Consumers' Attitude: A New Italian Perspective

**Bianca Polenzani** [1,*] **, Chiara Riganelli** [1] **and Andrea Marchini** [2]

1   Department of Economics, University of Perugia, Via A. Pascoli, 20 06123 Perugia, Italy; chiara.riganelli@gmail.com
2   Department of Agricultural, Food and Environmental Sciences, University of Perugia, Borgo XX Giugno, 74, 06121 Perugia, Italy; andrea.marchini@unipg.it
*   Correspondence: biancapolenzani@gmail.com; Tel.: +39-3405373932

**Abstract:** Extra virgin olive oil sustainability shows that it is strictly connected to local production and certifications. However, consumers' behaviour toward a local product is tied with the information exchange between producers and consumers. The aim of the research is to analyse, using a logistic regression model, if the attitudes, habits, and behaviours of the consumers influence their opinion on the sustainability of local extra virgin olive oil, relative to the three dimensions of sustainability (environmental, socio-cultural, and economic). This study tries to combine studies about agri-food sustainability and consumers' behaviour about local consumption, in an attempt to evaluate their attitude towards an agroecology food regime. Indeed, this information can be useful in order to plan adequate and specific interventions to improve the sustainability of the extra virgin olive oil production. The results indicate that the opinion about the sustainability of local extra virgin olive oil is linked to the information exchange between producers and consumers. This confirms that local alternative trade channels require numerous interventions in order to simplify and improve such exchange, both from a managerial and political point of view. Moreover, it can be useful to develop the information and communication technologies, in order to ensure the reliability, the transparency, and the security of the information exchange. This can also be useful to prevent frauds that are very common for this product.

**Keywords:** behaviour; sustainability; marketing; consumer; EVOO

## 1. Introduction

The term "sustainability" has different definitions, depending on the context in which it is used. Usually it describes environmental objectives only, but it could incorporate environment, economy, and society into a construct that forms the "three pillars of sustainability" [1,2]. Especially the last two aspects, which are often related to the local production and consumption. In Italy, where this research takes place, local food purchases are increasing, and they are considered a more sustainable alternative to the highly resource intensive modern agri-food supply chain [3–6], but they can also be in a position to enhance the sustainability in terms of socio-economic equity and environmental and local development [7–9].

The consumer behaviours from various regions of Italy are different, especially regarding food, traditions, and food production [10]. Especially for extra virgin olive oil (EVOO), the regional cultural identity is very important, even if it is widely considered one of the most important foods to improve health, and it is a pillar of the Mediterranean diet [11]. The aim of this research is to analyse

which attitudes, habits, and behaviours about local EVOO, influence the opinion of consumers on the sustainability of its production, relative to the three dimensions of sustainability (environmental, socio-cultural, and economic). Briefly, the research question is:

RQ. Which consumption habits, purchase choices, and behaviours affect the consumers' opinion about:

1. The environmental sustainability of local extra virgin olive oil production?
2. The cultural and social sustainability of local extra virgin olive oil production?
3. The economic and ethical sustainability of local extra virgin olive oil production?

Past literature has addressed environmental issues [12], the relation between regional production and quality perception [13], or social aspects and cultural issues related to the local production of olive oil and tourism [14]. Regarding the Italian EVOO, most studies focus on the link between quality perception, sustainability, and certifications, such as Protected Designation of Origin and Protected Geographical Indication (PDO and PGI) [10,15–17]. All these show that certifications and local production are linked to a perception of superior quality, creating a virtuous circle for sustainability. The current study moves in a different direction, trying to evaluate if and which behaviours and attitudes towards EVOO affect the consumers' opinion about the sustainability of this product and its production. In fact, consumption is frequently the result of an individual's own behaviour, in relation with that of other people [18]. Therefore, consumers' attitude can be a useful predictor of their possible choices, and how they address the relation between sustainability and local consumption. In this work all the declinations of sustainability (environmental, social, and economic) for EVOO will be discussed, as well as for other local products, with several considerations on consumers' interest in these aspects in relation to this product. This study also tries to shed a light on which purchase characteristics of Italian consumers are connected with the knowledge of EVOO's sustainability. This can be useful in order to evaluate how policies can develop it and increase local consumption of EVOO, but also how the producers could use this information to enhance their product. Furthermore, the results may be useful to evaluate whether technological innovation in supply chain management could be a strategic tool for the development of local consumption of extra virgin olive oil.

## 2. Literature Review

Among the policies, which are the pillar of the agri-food sustainability development, it is possible to distinguish two different paradigms [19]: the agro-industrial paradigm and the integrated territorial paradigm. The first one has its roots in the principles of the agricultural modernization [20] and leads to principles and policies of modernization, standardization, and globalization [21]. The second represents a rupture with the first one, because it aims to enhance the capacity of agri-food systems to promote specific territorial resources, integrated with nature conservation, tourism, and education [22–24]. Previous studies [25] tried to shed a light on which one of these two innovation paradigms had to occur to improve food system sustainability: environmental-corporate food regime (ECFR) and/or agroecology food regime (AEFR). The latter is tied to the socio-cultural aspect of sustainability, and it has the possibility to succeed in contexts and conditions in which the concept of sustainability becomes deeply rooted in culture and society [25]. This is the baseline of another question which the present work tries to answer: "Is the Italian market a context in which these conditions are present?"

Anyway, this paradigm arises also from a common need of change. In fact, according to transition theories [26–29], the economic activity is strictly embedded into relatively steady socio-technical systems [29], governed by regimes (i.e., coherent systems of rules and norms). Starting from that, the innovation contributes to the stability of existing socio-technical systems. When there is a crisis of the existing socio-technical system, the innovations are not able to give appropriate solutions. Therefore, a need for path-breaking innovation emerges. "Niches" (i.e., socio- technical systems which experiment with radically different cognitive frames, resource bases, and relational patterns [29]) can help the

system, creating the needed diversity and providing political, economic, and environmental solutions. Nurturing niches can be a powerful policy to strengthen processes of societal change "from below" [29]. Therefore, it is worth studying consumers' interest in local food, because it is one of the first drivers for the success of this paradigm of sustainability. Indeed, traditional and local products, which are usually produced with environmentally sustainable techniques, can be a useful tool in the promotion of socio-economic development, enhancement of territories, and biodiversity preservation [30].

EVOO is one of the most traditional Italian products, and its local production creates a virtuous circle, both for social and economic sustainability [15]. Economic conditions for sustainability refer to supporting the viability of local economies, the capacity to improve producers' incomes, and the quality of life. The local production and consumption thus become important sustainability tools, also due to the significance they have among consumers: the interest for the region, the tradition, and socio-economic welfare [15,31,32]. In fact, environmental sustainability is not the only concept on the basis of this issue, but cultural identity, food heritage, and rural integration are also taken into consideration [33–35]. The present literature review consists of three subsections: In the first, the link between local food consumption and sustainability is illustrated. In the second, new trends in Italian behaviour towards food and food culture, and how it influences food consumption and purchase is discussed. In the third subsection, a framework on the sustainability of local extra virgin olive oil is introduced. Finally, a section on new communication and information technologies is introduced. This is an important element because it is able to change the relationship between the companies in the supply chain, but also the relationship between producers and consumers. Indeed, information technologies can be an important strategic tool in the development of sustainable local products.

## 2.1. Local Food and Sustainability

The sustainability of rural areas, both socio-economic and environmental, has been more and more connected with the local products [36–38]. It has also become an important part of the tourist system [38], stimulating agricultural activity, creating job opportunities, encouraging entrepreneurship, reinforcing brand identity, and building cultural identity connected with food and related culture [39–42]. Indeed, consumption has become a tool to express a status, a state of mind, and personality [43]. In this context, local food consumption seems to reflect the growing need for participation in the production, quality, and traceability of food [44,45]. Furthermore, the reduction of the intermediaries between producer and consumer leads to a closer relationship between these two, that gives to the first one the possibility to control information towards final consumers related to the ethical and social values of the process [32]. In this way, it is simpler for the consumers to associate production with territory and society [7]. All considered, local food could be an answer to the increasingly not-so-sustainable global food supply [46]. This can be appreciated both by the consumers and by the producers, reconnecting them, and connecting the first one with the "real and sustainable food" dimension [47,48]. This dimension can be summarized in three main arguments [49] which are used both for the promotion and localization of food production and consumption: the environmental issue (therefore the reduction of impacts due to shorter distances), the possible reduction of environmental degradation, the enhancement of local producers, and the greater sense of community due to the "contact", direct or indirect, that could be created between producers and consumers through local networks [50].

By understanding the real cost of food production, and appreciating the producers' work, consumers may be more inclined to pay a premium price [17,51–53], which in turn allows producers to receive a dignified income for their work [54–57]. That is an important aspect of the'sustainability, because it allows to preserve both local small economic activities and cultural identity [5]. Local consumption can also lead to a renewal of small regional economies [54] and to ethical and health benefits (e.g., social responsibility, seasonality of production, promoting food safety, etc.) [32]. For what may concern environmental sustainability, large-scale food systems have decreased in soil and water quality and have negatively impacted climate change [58]. Agriculture

is one of the major contributors to the human impact on Earth's ecosystem, especially agriculture intensification, which is also insufficient to improve the socio-economic conditions of farmers [59]. The preservation of local agro-biodiversity may take advantage of the typical/local/traditional products [30]. In fact, local alternative approaches include farmers' markets, community-supported agriculture, box schemes, cooperatives, farm shops, and other initiatives [60]. These could be useful in order to face sustainability challenges, which the "conventional" food systems (i.e., characterised by large-scale, highly mechanized, and industrialized long food supply chains) cannot address [61].

## 2.2. New Trends in Italian Behaviour Towards Food

Nowadays, the consumers' behaviour about food has changed, and it keeps on changing. In the past, the most important requirement was "safety", but subsequently the need for "quality" has arisen, with the best ingredients and processes. Later the new trend was functional food, with a particular production process, rich in some elements, and poor in others. Today the attention is switching from the origin itself of the product, to the sustainable aspect of the origin and to its "emotional value". The exponents of this new trend express the willingness to reshape purchasing–consumption practices, re-building the entire system of production–distribution–consumption, founding it on shared principles of solidarity, sustainability, and equity [29,62]. The diffusion of this trend is widely helped by the high level of connection between consumers. In fact, the more the consumers are connected to each other, the more the trend becomes "social", and the more the innovation is shared, the more it becomes part of a common frame [29]. Because of the direct relationship between consumption and production paths, this change leads to the modification of the latter [29]. Therefore, the production and distribution systems have to be reshaped, in order to be aligned with this new sustainable trend.

However, during this process, consumers try to solve several problems and "consumption dilemmas", especially between the growing need of sustainability and the convenience to which they are used to [29]. The way consumers deal with them depends on the different basis of their values [63–70]. In order to face them, a need for new knowledge has arisen from both consumers and producers [70–73]. In particular, the producers had to change their production patterns, their ingredients, and their distribution channels, in order to satisfy consumers' requirements in terms of quality, safety, and sustainability. In fact, the relationship with reflexive consumers has become essential, because it can lead to several opportunities, and not just on the economic side. Nevertheless, consumers' knowledge should not be taken for granted, nor its value underestimated, but it should be curated and improved. In this way it is possible to create a bi-directional information flow where communication, negotiation, and a learning process on perceptions and concerns could take place [74,75]. By understanding the real cost of food production, and appreciating the producers' work, consumers may be more inclined to pay a premium price, which in turn allows producers to receive a dignified income for their work. That is an important aspect of the sustainability, because it allows the ability to preserve both economic activities and cultural identity. Local consumption can also lead to a renewal of the small regional economies, and also to ethical and health benefits (e.g., social responsibility, seasonality of production, promoting food safety, etc.). In fact, new alternative networks need this cognitive aspect, especially in their organisational design [76].

This local-based new food paradigm is not only an expression of consumers' choices within a given supply of partial options [77]. It represents an effort to build a new system, based on environmental, social, and economic sustainability. In this new system there is a shared and coherent set of meanings, principles, and goals, there is communication, which translates into consumption choices. Through that, new knowledge systems, skills, rules, and organisational patterns can be developed. This dialectic between niches and the regime can lead to a radical change in food culture and practices, in terms of consumers' demand, social customs, and habits. This is intrinsic in the nature of food, which conditions both everyday life and more complex social dynamics at the local and global level [78]. To appreciate it, there is the need of knowledge, from both the producers and the consumers. There should be an interaction between different sources of knowledge, for both

the producers and the consumers. In all these issues, targeted policies can be a useful strategic tool in developing and improving these processes. In fact, policies affect the production process, the practices involved in food distribution, and consumption. They can enhance the "quality" through certifications, interventions, codifications, etc. They can also find appropriate solutions to the specific needs of the local producers.

### 2.3. Local Extra Virgin Olive Oil and Sustainability

In Italy, although extra virgin olive oil is perceived as a pillar of the traditional Italian diet, the convictions, knowledge, and cultural background vary widely from region to region, and obviously from consumer to consumer. For what may concern the environmental sustainability of EVOO, a multidisciplinary study was performed by Proietti et al. [79], about the carbon foot print of this product, assessing that EVOO production, under certain conditions, mainly local origin and traditional practices, could be a potential carbon negative product [79]. An important issue about local EVOO is related to the condition of economic sustainability for the consumer, which derives from the willingness to pay a premium price for the products of the territory [80,81]. This is true especially for the modern consumer, who is characterised by a greater sensitivity to quality, in terms of organoleptic properties, healthiness, and respect for the natural and cultural environment of the production territories [82–84].

In terms of connection between quality perception and sustainability, all the studies formerly mentioned show that certifications and local production are linked to a perception of superior quality, creating a virtuous circle for sustainability. The current study tries to evaluate whether consumption habits and purchasing choices throughout the country are related to the opinion about sustainability of local EVOO, in order to understand how this issue is perceived and faced by Italian consumers, and which instruments can be used to develop it. This is important to evaluate how policies can be more useful to develop the sustainability and to strength local consumption of EVOO. In order to study this topic, it can be useful to explore the complexity of the relation between "local food" and consumers. Previous research has tied the concept of "local food" with the geographical and cultural conditions, which lead to the starting up of local food networks [85–93]. Another group of studies have focused on the consumers' perception towards local food, related to the quality and marketing of products [17,52,94–96]. Brunori (2007) shows that the origin of food products was one of the most recurrent factors in relation to the concept of quality [89], and he suggests a distinction between "local food", "locality food", and "localist food". The term "local" is referred to a short-distance relationship in a community, which is tied by the same food habits and traditions [85,89]. "Locality food" is focused on the particular origin of the product, giving less importance to the "community" [85,89]. The concept of "localist" is not related to a traditional food habit, but it is a product deliberately chosen from a set of products by consumers living in the same place, and this concept implies consumers' willingness to reconstruct local identities by the regular consumption of food products, although they do not belong to the rural traditions of that local area [85,89].

Italy is a particular case, because there are several different food traditions, which have an important historical value in different regions. The concept of "local" is usually tied with regional and national boundaries [97–100]. Another interpretation of "local" concept is related to the food traditions from a certain place [101], or to the symbolic and cultural value of a food product [102]. However local products themselves are often tied with a particular territory or climate, because their characteristics and production processes, which lead to the best food products that can be obtained. Thus, food consumption is not only the satisfaction of a functional need, but it is also a tool which satisfies a social and political need [103].

Taking this into account, extra virgin olive oil can be considered a local and a localist product. There is a widely superficial knowledge of it, because it is a common product which is used every day and it does not stimulate the consumers' curiosity in terms of taste, composition, characteristics, etc. This product is not so customisable, due to its production method and environmental characteristics, which are the most important factor in the extra virgin olive oil's characterisation. It is a simple product,

and every region and/or territory has its traditions and preferences relative to the organoleptic characteristics of this product (taste, colour, etc.). These are important factors, which should be considered in order to enhance the characteristics of local extra virgin olive oil. In fact, this product to be fully appreciated requires a particular knowledge both form the local producers and from the consumers. Local niche production has an important and difficult objective: assure the safety and quality standards, while also remaining different. The local extra virgin olive oil should distinguish itself by being different from the other commercial extra virgin olive oils, but the process must reach the same standards in quality and in safety. On the other hand, the difficulties faced by producers should be recognized and awarded by consumers. In fact, this paradigm needs the knowledge in input, from the producers, and in output, from the consumers, whom should be widely informed in order to pay a premium and right price for the product. In the final analysis, the study tries to evaluate where the attitudes, behaviours, and characteristics (i.e., the propensity to be informed and conscious) would have an effect on the consumers' opinion on the sustainability of this local production. This knowledge can be used to determine how the information strategy can contribute to include this production in a virtuous circle of overall sustainability. These are important elements to evaluate, because it can influence policies making them more useful in the promotion of sustainability through the consumption of local products. This information can also be useful for the producers, whom could use it to better promote their products.

## 2.4. Information and Communication Technologies and Local Products

Although the information exchange through the traditional communication channels is important, consumers today also seek for a rational support to their purchase choices [104]. A better system of traceability and identification can be an interesting answer to this and to other consumers' needs [105]. A considerable option, which can support and develop both local production and consumption is the Blockchain technology. In fact, this can improve the communication, the coordination, and the cooperation between the nodes of the supply chain (including consumers) [106,107] by developing a sustainable agriculture supply chain [108]. This technology can be used to improve the monitoring of both socio-economic and environmental sides of the processes [107]. In fact, it can facilitate and/or amplify controls, which can help prevent different types of fraud (see, for example, the cases of "Italian sounding effect"and fakes on the origin of the products) [109,110]. For what may concern the environmental aspect, several studies focused on the use of new information and communication technologies (ICT) in the circular economy and in the reduction of food waste [111,112].

Blockchain is a distributed and immutable database (with limited use for owners of encrypted access keys). A possible future adoption of this technology cannot disregard the use of intelligent devices capable of recording data, preferably in real time, and subsequently collected in a single distributed database, which is transparent, accessible, and which does not depend on trust in a central authority [113]. From this perspective, the blockchain represents an improvement of modern technological systems for collecting and sharing information, since it would allow, by using the IoT (Internet of Things), to gather the information obtained through these devices, thus becoming their collection infrastructure. In this way the increasing consumers' demand for information can be satisfied by improving the sustainability and the efficiency of the supply chain [107]. In fact, new ICT could increase consumers' trust towards food products and producers [114].

For what may concern extra virgin olive oil, as previously said, a traceability system based on blockchain can be useful in order to develop the actual control systems, including certifications [115,116]. In fact, it would be possible to trace the product with more efficiency and versatility, and this could lead to a better supply chain transparency and better anti-fraud systems [115]. One of the main concerns in adopting this technology is consumer acceptance and interest in it. In fact, the attention to food safety justifies an increasing willingness to pay for food traceability [117,118]. However, the knowledge of the product (in terms of quality, safety, and certifications) is important in order to determine the consumers' "willingness to pay" for a food product with traceability systems

which integrate new ICT [117,119,120]. The blockchain could be an important tool especially for certifications [121]. In fact, the compliance with regulations, compatible with the speed of change and/or the updating of the same, represents a significant problem for companies, especially if operating in international areas. The ease of forgery risks undermining these relationships. With blockchain technology, one could easily overcome this problem [122]. However, the cost of implementation is very high, especially for small local producers [115]. Furthermore, the technological baseline required for the implementation of blockchain is expensive and advanced (for example: smart devices for tracking and tracing, enterprise resource planning softwares, etc.). Moreover, an important barrier to adoption for small producers is represented by the inefficiency of the internet (or a too slow connection) and by the delay in adoption, in other words, the impossibility of simultaneous adoption by all the actors operating in the chain. In addition to this, even the scarce digitalization of the partners could worry a company at the time of adoption, since it would make it difficult to use the blockchain even internally within its business [122]. Small and traditional enterprises, especially, may be a familial business (which are very common in Italy for EVOO), could have weak incentives to implement such an innovative technology. However, these companies could reduce their implementation costs with a collective strategy and with an institutional arrangement by the farmers [31,107,123]. For example, Scuderi et al. [121] aims to study an operating model that exploits the blockchain for a product obtained from the transformation of "Arancia Rossa di Sicilia PGI", such as blood-orange juice. The analysis on their part has highlighted how a platform through which the PDO and PGI producers have registered and interfaced is actually a complex goal [121]. Indeed, there is the need to create an integrated and transparent food supply chain rich in information that guarantees both food safety and the ethics of production, as well as the safety and safeguarding of the consumer for a complex product [121].

## 3. Materials and Methods

In order to collect data, an online questionnaire was used (made with Google Form), which has been spread through the Internet, using a virtual snowball sampling technique, which is a non-probability sampling technique where research participants recruit other participants for the study. The use of the Internet was important, because it is able to reach people from different geographical areas [124]. In fact, the questionnaire was diffused using the main social networks in particular, Facebook and Instagram. On the first network, the respondents were requested to share the questionnaire one time before the completion. On the second network, the questionnaire was spread by two influencers (people that work on social networks and which have a large audience), who encourage their followers to re-share it one time. The data has been gathered in Italy between March and April 2019. The sample contains 347 respondents, and it is suggested that a number of 300 and above is an acceptable sample size, ensuing valid statistical estimates and reliable results [125,126]. The database created contains cross-sectional data in which the statistic units are the respondents and each variable refers to a specific question in the questionnaire. First of all, the questionnaire contains a question about the region of origin. Successively, there is a screening question asking about the consumption of extra virgin olive oil (i.e., if they are a consumer or not) (This question is necessary in order to consider only consumers of EVOO. As a matter of fact, consumers who never consume EVOO could not be allowed to answer the questionnaire.). Then interviewees were asked, by using a Likert-scale from 1 (never) up to 7 (very frequently), the frequency with which they buy EVOO from producers. Furthermore, the respondents were asked if he/she is responsible for the purchase or if they are the producer of the EVOO consumed. These are the questions about consumption habits for EVOO. Interviewees were also asked if they have ever participated in events regarding EVOO (a closed question). Moreover, interviewees were asked in the questionnaire to express how much importance they give to some attributes of EVOO at the purchase moment, by using a Likert-scale from 1 (not important) up to 7 (very important). These questions were done in order to investigate the behaviour towards this product. These features were related with both quality aspects (i.e., local origin, certifications, taste, and nutritional properties), and visual or market aspects

(i.e., brand, sales, and packaging). In particular, packaging aspects considered were: the colour of the bottle (and specifically the possibility to see the product through it), dimension of the package, and material.

Both of these two categories of information, are very important for extra virgin olive oil, since they can lead to an assessment of healthiness, and therefore quality [127–130]. The interviewees were also asked if they consider themselves experts of EVOO (being an expert usually determines further knowledge of its sustainability), and the importance of friends' and relatives' advices.

In relation with consumption habits, in the third part of the questionnaire, the interviewees were asked to express their frequency of purchase of big brands and private label olive oils, both numerous in Italy (although most of the private labels in Italy are produced by big brands). The question was formed by using a 7 point Likert-scale. All of this information represents the independent variables in the analysis.

Considering the previous studies and the specificities of the segment object of the analysis (the extra virgin olive oil) we asked interviewees to:

"Please indicate if you agree or disagree with the following statements:

The production and consumption of local Extra Virgin Olive Oil help to preserve:

1.  The environment;
2.  Traditions, and the local culture;
3.  The viability of local small producers."

These three statements were made in order to investigate the consumers' awareness about the sustainability of local extra virgin olive oil. They are the dependent variables of our analysis related to their behaviour. These are closed questions, because of the nature of the analysis. Finally, some social and demographic characteristics were collected in order to verify the results about the opinion on the three declinations of sustainability. Considering this last set of variables, we have a sample of respondents that were 68% female and 32% male, most of them in the age group 18–45.A total of 51% of respondents have a higher education (a degree), followed by 41%, which have completed high school. For what may concern the income bracket, 56% of respondents are in a low-income bracket, while 34% are in the mid-range income bracket. Finally, 35% of respondents have a household size consisting of 4 people, followed by 22% with a household size consisting of 3 people. In order to test the research question we analysed the data by the following equation obtained by consumer $i$ for each label information $l$:

$$Y_i l = \sum_b \gamma_b HABITS_i + \sum_k \beta_k BEHAV_i + \sum_j \theta_j SOC_i + \epsilon_i \tag{1}$$

where $Y_i l$ denotes the outcome variables. These are dummy variables that show whether or not consumers are in agreement with the statements about the three declinations of sustainability. $HABITS_i$ indicates the habits about EVOO's consumption and purchase. $BEHAV_i$ represents the set of characteristics through which purchase choices are made, and which denotes the respondents' behaviour towards the EVOO. $SOC_i$ represents the socio-demographic characteristics of the sample. The set of variables include *gender*, *age*, *size of household*, level of *education*, and of *household income*. It is used in order to control the results of BEHAV., the independent variables that represent the respondents' behavior towards local Extra Virgin Olive Oil. Table 1 shows descriptive statistics of the variables used in the analysis.

Considering the nature of the dependent variables, we estimated using a logistic model [131,132]. The analysis was performed using Stata Software 13. The model also includes the robust estimation of standard error in order to avoid possible serial correlation, and it takes into consideration the violation of the normality assumption and the presence of outliers [133]. Furthermore, for every model estimated in the analysis, the indices of fitting are considered: every model has a considerable reliability, because for each one we can reject the null hypothesis of a combined variables effect equal to zero (Wald

chi-square statistic). For the interpretation of the result, the odds ratio (OR) is shown. Instead of performing the marginal effect, in this paper we have chosen to perform the odds ratio estimation, that gives us information about the event probability of the dependent variables in relation to the value assumed by the explanatory variables.

**Table 1.** Definition of variables, means, frequencies, and standard deviations.

| Variable Name | Description | Obs | Mean | SD |
|---|---|---|---|---|
| **Dependent variables** | | | | |
| Environmental Sustainability | Dummy, respondents in accordance=1; otherwise=0 | 340 | 0.39 | 0.49 |
| Social Sustainability | Dummy, respondents in accordance=1; otherwise=0 | 343 | 0.87 | 0.33 |
| Economic Sustainability | Dummy, respondents in accordance=1; otherwise=0 | 343 | 0.79 | 0.41 |
| **Independent variables** | | | | |
| Habitual EVOO consumer | Dummy, habitual consumers=1; otherwise=0 | 345 | 0.98 | 0.13 |
| Responsible of the EVOO purchase | Dummy, respondents responsible=1; otherwise=0 | 337 | 0.55 | 0.50 |
| Producer of the EVOO consumed | Dummy, respondents producers=1; otherwise=0 | 345 | 0.33 | 0.47 |
| Participation to events regarding EVOO | Dummy, respondents who participated=1; otherwise=0 | 344 | 0.50 | 0.50 |
| Frequency of purchase EVOO from producers | Scale from 1 to 7 | 335 | 4.83 | 2.32 |
| Consider themselves EVOO experts | Scale from 1 to 7 | 343 | 4.28 | 2.12 |
| Brand | Scale from 1 to 7 | 335 | 4.54 | 2.13 |
| Local | Scale from 1 to 7 | 341 | 6.15 | 1.40 |
| Certifications | Scale from 1 to 7 | 340 | 5.72 | 1.57 |
| Taste | Scale from 1 to 7 | 341 | 6.51 | 0.88 |
| Nutritional quality | Scale from 1 to 7 | 341 | 5.95 | 1.41 |
| Price | Scale from 1 to 7 | 341 | 4.82 | 1.57 |
| Visual aspect of the packaging | Scale from 1 to 7 | 341 | 3.22 | 1.88 |
| Material | Scale from 1 to 7 | 341 | 4.85 | 1.91 |
| Big size | Scale from 1 to 7 | 337 | 4.26 | 1.89 |
| Small size | Scale from 1 to 7 | 336 | 3.22 | 1.74 |
| Promotions and sales | Scale from 1 to 7 | 336 | 4.26 | 2.04 |
| Possibility to see the product | Scale from 1 to 7 | 340 | 4.65 | 2.07 |
| Importance of friends and relatives advices | Scale from 1 to 7 | 342 | 4.73 | 1.82 |
| Habitual consumption of private labels EVOO | Scale from 1 to 7 | 340 | 2.15 | 1.59 |
| Habitual consumption of big brands EVOO | Scale from 1 to 7 | 340 | 2.94 | 2.01 |
| SOC - Woman | Dummy, respondents are woman =1; otherwise =0 | 343 | 0.69 | 0.46 |
| SOC - Age | Dummy, respondents are 18-45=1; otherwise =0 | 343 | 0.60 | 0.49 |
| SOC - Household size | Scale from 1 to 5, household size of respondents | 344 | 3.21 | 1.22 |
| SOC - Education | Scale from 1 to 3, level of education of respondents | 343 | 2.44 | 0.63 |
| SOC - Household income | Scale from 1 to 3, level of household income | 329 | 1.58 | 0.77 |

## 4. Results

To test the research question previously formulated through Equation (1), we performed a set of three Logit models, one for each statement on sustainability, which represented the dependent variables of the Table 2.

RQ. Do consumption habits, purchase choices, and behaviours affect the consumers' opinion about:

1. The environmental sustainability of local extra virgin olive oil production?
2. The cultural and social sustainability of local extra virgin olive oil production?
3. The economic and ethical sustainability of local extra virgin olive oil production?

### 4.1. Environmental Sustainability

Considering environmental sustainability, two factors show positive and significant impact: "visual aspect of the packaging" and "consider themselves EVOO experts". The importance given to the visual aspect of the packaging can be linked to the information. In fact, for extra virgin olive oil, packaging is an important information tool because producers in Italy have to provide lots of information through it, among which are the origin, the presence of certifications, the production building, etc. Therefore, the information provided through the bottle can be an element connected to consumers' sustainability awareness. On the other hand, consumers who consider themselves experts of this product can probably appreciate all the organoleptic shades of an artisanal extra virgin olive oil. These are mostly attributable to the different olive tree varieties. Having a deep knowledge of the product and the production process can lead to the awareness of its sustainability. Therefore, consumers who are experts of this product know that the environment and the variety of olive trees are enhanced by the local production of extra virgin olive oil. Therefore these consumers may be more aware of the environmental sustainability of this production.

**Table 2.** Model estimation: beta coefficients and odds ratio for voluntary label disclosure.

| Variables | (Model 1) Environmental Sustainability | | | | (Model 2) Social Sustainability | | | | (Model 3) Economic Sustainability | | | |
|---|---|---|---|---|---|---|---|---|---|---|---|---|
| | *β* | *SE* | *OR* | *SE* | *β* | *SE* | *OR* | *SE* | *β* | *SE* | *OR* | *SE* |
| Habitual EVOO consumer | 0.22 | [1.04] | 1.25 | [1.30] | 1.79 | [0.92] | 6.00 | [5.50] | 0.74 | [1.08] | 2.09 | [2.27] |
| Responsible of the EVOO purchase | 0.54 | [0.34] | 1.71 | [0.59] | 0.31 | [0.53] | 1.37 | [0.72] | 0.06 | [0.44] | 1.06 | [0.46] |
| Producer of the EVOO consumed | 0.34 | [0.33] | 1.41 | [0.46] | 0.28 | [0.56] | 1.32 | [0.74] | −0.06 | [0.44] | 0.95 | [0.42] |
| Participation to events regarding EVOO | −0.02 | [0.31] | 0.98 | [0.30] | 0.23 | [0.43] | 1.26 | [0.55] | 0.13 | [0.38] | 1.13 | [0.44] |
| Frequency of purchase EVOO from producers | 0.02 | [0.07] | 1.02 | [0.08] | −0.04 | [0.10] | 0.96 | [0.09] | 0.15 | [0.08] | 0.16 | [0.10] |
| Consider themselves EVOO experts | 0.19 ** | [0.08] | 1.21 ** | [0.09] | 0.04 | [0.11] | 1.04 | [0.12] | 0.05 | [0.09] | 1.05 | [0.10] |
| Brand | 0.10 | [0.07] | 1.11 | [0.08] | 0.02 | [0.11] | 1.02 | [0.11] | −0.02 | [0.08] | 0.98 | [0.08] |
| Local | 0.15 | [0.14] | 1.17 | [0.16] | 0.18 | [0.13] | 1.20 | [0.16] | −0.05 | [0.12] | 0.95 | [0.11] |
| Certifications | 0.04 | [0.11] | 1.04 | [0.11] | 0.29 ** | [0.11] | 1.34 ** | [0.15] | 0.34 *** | [0.10] | 1.41 *** | [0.15] |
| Taste | 0.29 | [0.19] | 1.34 | [0.25] | 0.29 | [0.22] | 1.34 | [0.29] | −0.05 | [0.18] | 0.95 | [0.17] |
| Nutritional quality | 0.21 | [0.16] | 1.23 | [0.19] | −0.05 | [0.14] | 0.95 | [0.13] | −0.01 | [0.11] | 0.99 | [0.11] |
| Price | 0.17 | [0.11] | 1.19 | [0.12] | 0.02 | [0.15] | 1.02 | [0.15] | 0.16 | [0.12] | 1.17 | [0.14] |
| Visual aspect of the packaging | 0.21 ** | [0.10] | 1.24 ** | [0.12] | −0.06 | [0.13] | 0.94 | [0.12] | 0.02 | [0.11] | 1.02 | [0.11] |
| Material | −0.02 | [0.09] | 0.98 | [0.09] | 0.07 | [0.12] | 1.08 | [0.12] | 0.00 | [0.10] | 1.00 | [0.10] |
| Big size | 0.04 | [0.08] | 1.04 | [0.08] | 0.00 | [0.12] | 1.00 | [0.12] | 0.11 | [0.10] | 1.11 | [0.11] |
| Small size | 0.01 | [0.09] | 1.01 | [0.10] | 0.02 | [0.12] | 1.02 | [0.12] | 0.14 | [0.10] | 1.15 | [0.12] |
| Promotions and sales | −0.09 | [0.09] | 0.91 | [0.08] | 0.09 | [0.11] | 1.10 | [0.12] | −0.21 ** | [0.10] | 0.81 ** | [0.08] |
| Possibility to see the product | 0.04 | [0.08] | 1.04 | [0.08] | 0.06 | [0.11] | 1.07 | [0.11] | −0.04 | [0.08] | 0.96 | [0.08] |
| Importance of friends and relatives advices | 0.08 | [0.10] | 1.08 | [0.11] | 0.22 | [0.11] | 1.25 | [0.14] | 0.14 | [0.09] | 1.15 | [0.11] |
| Habitual consumption of private labels EVOO | 0.18 | [0.11] | 1.20 | [0.13] | 0.03 | [0.16] | 1.03 | [0.17] | 0.25 | [0.15] | 1.29 | [0.19] |
| Habitual consumption of big brands EVOO | −0.16 | [0.09] | 0.85 | [0.07] | −0.14 | [0.11] | 0.87 | [0.10] | −0.17 | [0.09] | 0.84 | [0.08] |
| SOC - Woman | −0.50 | [0.32] | 0.60 | [0.19] | 0.51 | [0.45] | 1.66 | [0.75] | 0.37 | [0.39] | 1.45 | [0.57] |
| SOC - Age: 18–45 | −0.08 | [0.37] | 0.92 | [0.34] | −0.06 | [0.52] | 0.94 | [0.50] | 0.40 | [0.40] | 1.49 | [0.60] |
| SOC - Household size: 2 members | 0.09 | [0.53] | 1.10 | [0.58] | −0.24 | [0.76] | 0.79 | [0.60] | −0.78 | [0.69] | 0.46 | [0.32] |
| SOC - Household size: 3 members | 0.16 | [0.51] | 1.17 | [0.60] | 0.06 | [0.75] | 1.07 | [0.80] | 0.71 | [0.73] | 1.07 | [0.78] |
| SOC - Household size: 4 members | −0.27 | [0.54] | 0.76 | [0.41] | −0.28 | [0.67] | 0.76 | [0.51] | −0.19 | [0.67] | 0.82 | [0.55] |
| SOC - Household size: >4 members | −0.64 | [0.64] | 0.53 | [0.34] | −0.28 | [0.83] | 0.76 | [0.63] | −0.10 | [0.75] | 0.91 | [0.63] |
| SOC - Education: High school | −0.38 | [0.55] | 0.68 | [0.37] | 0.15 | [0.63] | 1.16 | [0.73] | 0.75 | [0.50] | 2.11 | [1.06] |
| SOC - Education: Degree | −0.46 | [0.55] | 0.63 | [0.35] | 0.21 | [0.63] | 1.24 | [0.78] | 0.52 | [0.50] | 1.69 | [0.84] |
| SOC - Income <36,151.98 | −0.39 | [0.33] | 0.68 | [0.22] | 0.17 | [0.48] | 1.19 | [0.56] | −0.47 | [0.42] | 0.63 | [0.26] |
| SOC - Income: from 36,151.98 to 70,000.00 | 1.13 | [0.69] | 3.10 | [2.15] | 0.27 | [0.90] | 1.31 | [1.18] | −0.07 | [0.80] | 0.94 | [0.74] |
| SOC - Income: from 70,000.00 to 100,000.00 | 0.08 | [0.70] | 1.09 | [0.76] | 0.43 | [1.18] | 1.53 | [1.82] | −0.13 | [0.95] | 0.88 | [0.84] |
| Observations | 300 | | | | 302 | | | | 302 | | | |

*** $p < 0.01$, ** $p < 0.05$, There are robust standard errors (SE) in brackets.

*4.2. Social Sustainability*

Taking into account the second declination of sustainability (social), only one factor shows a positive and significant impact: "certifications".This result can be linked with the nature of the most common EVOO certifications. In fact, most of the certification is related to local production and safeguard of traditions (such as PDO and PGI). The bond between origin and safeguard of traditions is an important element in a lot of certifications. Therefore, it is comprehensible, because the interest in certifications is positively related to the cultural and social sustainability of local extra virgin olive oil. This result can be related to both the information exchange (between producers and consumers) and the origin of the product.

*4.3. Economic Sustainability*

Finally, "certifications" also show a positive and significant impact on opinions about economic sustainability. "Presence of promotions and sales" show a significant and negative impact, therefore consumers who buy cheap or discounted extra virgin olive oil have a greater probability to think that the production and consumption of extra virgin olive oil does not help to preserve the viability of local small producers. As said above, certifications are mostly related to the tradition and the origin of the product, and they enhance the local origin. Therefore, consumers who are interested in certification may have an interest in the economic sustainability of extra virgin olive oil. This concept is linked with the negative impact of interest in promotions and sales. Indeed, consumers who seek discounted products are less likely to consider economically sustainable EVOO production, because local and niche products are usually not discounted due to their value.

Although the significances are few, they are quite important for the topic of the research. In fact, the consumers' who are aware of the sustainability of local EVOO production are informed and interested in information. Certifications for this product are one of the most important means of communication, within the label (i.e., the bottle) where they can be found and recognised. Therefore, the significance of these elements can highlight a connection between the means for enhancing quality (label and certifications) and sustainability awareness. Furthermore, provide information seems to be a crucial elements in the safeguard of small local producers of extra virgin olive oil. Indeed, if the information were more accessible, consumers would likely be willing to pay a fair price for local products, which can reward producers [80].

## 5. Discussion

The results show that consumers who are in agreement with the proposed statements are consumers who pay more attention to information rather than to tangible aspects. Therefore, this confirms what was found by the previous studies: there is a connection between knowledge, information, and awareness of sustainable local consumption, confirming previous studies [51–53, 74,75]. Furthermore, an exchange of information, between producers and consumers, is important in order to be recognized by the consumers as a sustainable product, rewarding so the small local producers [51–53,74–76,89].

In fact, consumers who are better informed about this product have a higher propensity to consider local EVOO an important product for the overall sustainability. The fact that consumers with better information can affect the opinion about the overall sustainability of extra virgin olive oil is important. In fact, previous studies have highlighted how the consumption of local traditional products, usually linked with traditions, heritage, and family can easily influence the opinion about the sustainability of this product for the place of production [14,101,102]. This heritage can lead to a greater knowledge of the product and of the production process, but nowadays it is not sufficient. This study was done in order to understand with which consumption features this this awareness is linked. From the previous studies it is widely known that consumers, for local products, are orientated towards certified ones [10,13,15,17,81,115,134]. However, the results of the present research

show that certifications also are related with consumers' awareness about sustainability. This confirms that the information about local products can be an important strategic tool which can lead to more conscious purchase choices.

Therefore, it can be important to develop the exchange of information between producers and consumers, making it more efficient. From previous studies it seems that consumers accept a blockchain system for extra virgin olive oil traceability [115]. The results of the present study follow the same line, showing how much the consumers' awareness depends on this information, and how certifications are an important part of it. Therefore, can be important to develop certification systems using blockchain technology. In fact, as previously said, this technology which provides transparent, safe, and better information, is important especially for small producers [115]. To develop consumer awareness through a better information system can make consumers more inclined to pay a greater price for a product transparently tracked with this technology. Furthermore, this system can make them more aware about the differences between a standard product and a local one. However, the greater problem with these technologies is the high cost of implementation, which is rewarded only if the production is quite big [115]. Furthermore, incentives for technological resources are scarce, especially for the sustainability of the supply chain [107]. On the other hand, if PDO and/or PGI producers would implement this technology in association form, informed and conscious consumers may pay a greater price for their product.

The main limitations of the present study concern the sampling which has generated a small number of respondents. Extending the research to a greater population could lead to more reliable results. Furthermore, the data gathering method used in the current study could lead to a censured sample. In fact, spreading the questionnaires through the Internet might exclude a part of the population who does not use social networks. Another limitation refers to the country (which is a peculiar case study for EVOO) and the product. It could be an interesting avenue to explore similar research with other products (such as milk or meat), or with the same product, but in other countries, for example Spain. In this case, it could be also interesting to make a comparison with Italy, because the big and intensive production of extra virgin olive oil in Spain is very different from the Italian one.

## 6. Conclusions

The study may be useful to understand which factors related to consumer behaviour influence the perception of extra virgin olive oil sustainability. This knowledge can be used by policy makers to plan actions in support of sustainability and its enhancement for the success of local business. In fact, knowing the consumers' opinion about the sustainability of this local product can be important in order to evaluate how EVOO's local production could be developed. The current study highlights the key-role of policies in developing certification methods, by maybe using new Information and communication technologies. This is important in order to make the information exchange better and more transparent, but also to avoid fraud [110]. This is a crucial problem in Italy, where certification systems are not so smart and efficient, and where fraud is very frequent, harming lots of local business [110].

Finally, the results highlight implications related to the management of the product. Local EVOO should not be sold by large-scale retailers, and should not be interested in sales and discounts, because these are elements that consumers do not expect from a local and sustainable extra virgin olive oil. Even if it seems to be a managerial implication, it is also political, because the development of local trade channels, and the enhancement of small, local production, should be taken into account by Italian and European agricultural policies. Information and knowledge can create awareness, which can increase the consumers' willingness to pay for local extra virgin olive oil [51–53]. This can be an opportunity for local producers. Therefore, it can be useful for small producers to implement a blockchain technology which could improve the quality of the information and the consumers' attitude towards certifications [115]. The union between technology, traditions, and heritage can not be only possible, but useful. It is also important to consider that, although integrated territorial paradigm

presents several weak points [25], it is worth studying and increasing the consumers' interest for the sustainability of local food. Indeed, making the consumers more aware about the sustainability of local productions is very important for the success of this paradigm. Taking into account the Italian background and possibilities, if this exchange of information will be developed, consumers could embrace this paradigm of sustainability for extra virgin olive oil.

**Author Contributions:** Visualisation, investigation, writing—original drafts, B.P.; methodology, C.R.; conceptualization, supervision, and project administration, A.M. All authors have read and agreed to the published version of the manuscript.

**Funding:** This research received no external funding.

**Conflicts of Interest:** The authors declare no conflict of interest.

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
