# Peer review of "Sustainability Perception of Local Extra Virgin Olive Oil and Consumers’ Attitude: A New Italian Perspective"

_sustainability, doi:10.3390/su12030920_

Round 1

Reviewer 1 Report

The manuscript entitled “SUSTAINABILITY PERCEPTION OF LOCAL EXTRA VIRGIN OLIVE OIL AND CONSUMERS’ ATTITUDE: A NEW ITALIAN PERSPECTIVE” presents an interesting issue, however it requires some amendments.

Abstract:

Abstract should be improved – the sentence like “The idea of this paper steams from previous works about agri-food system...” should be omitted. Authors should avoid the narrative language and should focus only on the main aspects of the study. A good abstract should include a problem statement, background, methodology, key findings and a conclusions, which assist the reader to understand the study.  The results and conclusions should be presented in this section. Line 13 – “PACS: J0101” – should be removed

Introduction:

Lines 21-22 – “The product object of the present research is Extra Virgin Olive Oil, which is a pillar of Italian and Mediterranean diet.” – should be removed from this section Line 23 – “In Italy each region has its own behaviour on food, traditions, sustainability, and food production” this sentence is confusing and it should be corrected. The consumer behaviors from various regions of Italy are differ not a region has a different behavior …. Lines 48-51 – this paragraph is redundant. Some sentences or some wording used by authors are confusing (e.g. “Previous researches” – is should be “Previous studies”) Authors should improve the language in whole manuscript. Line 90 – “food[35–37].” Typos – it should be “food [35–37]”. Line 174 – “Our research tries to evaluate” - Scientific writing has traditionally been third person, passive voice – authors should avoid first person.

Materials and methods:

Authors should explain precisely how the “Snowball sampling technique” was applied via google form. Was the link to the survey sent via email (what was the database for email) or was it posed on some webpage (of university or personal Facebook). How did authors encourage respondents to pass along the link with the survey? The number of respondents is small (taking into account that this is not a specific samples with specific inclusion criteria but rather random). Power of samples size must be presented and sample size must be justified as well as it must be indicated in limitation section as a potential bias. The question form applied in the survey should be presented. Taking into account the fact, that authors tried to analyze not only consumer behaviors but habits, the proper tool should be presented. The p<0.10 – as a not significant (but tend to be significant) should be omitted due to low samples size (and resultant high risk of bias)

Results:

Results should be presented in clearly in paragraphs (e.g. as a summarizing the meaning of the result – not only as a repetition of the results presented in tables). Some words should be corrected (e.g., consumer think – We do not know what they think we just could say what they declare, therefore authors should use “consumer declared think” )

Discussion:

The discussion section should be broaden. Authors should relate the findings to those of similar studies and point out the differences and similarities between the studies. Authors should add the appropriate references in this section. The limitation section should be expanded (for the sample size, data gathering, etc.)

Conclusion:

This section must be improved. Authors should present the conclusions of the study not the other information.

The manuscript must be corrected according to the authors guideline!

Author Response

Abstract should be improved – the sentence like “The idea of this paper steams from previous works about agri-food system...” should be omitted. Authors should avoid the narrative language and should focus only on the main aspects of the study. A good abstract should include a problem statement, background, methodology, key findings and a conclusions, which assist the reader to understand the study.  The results and conclusions should be presented in this section.

Thank you for this comment. By following your suggestion, we have improved the abstract by including all the elements recommended.

Line 13 – “PACS: J0101” – should be removed

Done

Lines 21-22 – “The product object of the present research is Extra Virgin Olive Oil, which is a pillar of Italian and Mediterranean diet.” – should be removed from this section

Done

Line 23 – “In Italy each region has its own behaviour on food, traditions, sustainability, and food production” this sentence is confusing and it should be corrected. The consumer behaviors from various regions of Italy are differ not a region has a different behavior ….

Thank you for this comment. We have reworded the sentence by following your suggestion.

Lines 48-51 – this paragraph is redundant. Some sentences or some wording used by authors are confusing (e.g. “Previous researches” is should be “Previous studies”) Authors should improve the language in whole manuscript.

Thank you for this suggestion. We removed the paragraph and we have improved the language in whole manuscript.

Line 90 – “food[35–37].” Typos – it should be “food [35–37]”

Done

Line 174 – “Our research tries to evaluate” - Scientific writing has traditionally been third person, passive voice – authors should avoid first person.

Thank you for this comment. We have removed the first person from the whole manuscript

Authors should explain precisely how the “Snowball sampling technique” was applied via google form. Was the link to the survey sent via email (what was the database for email) or was it posed on some webpage (of university or personal Facebook). How did authors encourage respondents to pass along the link with the survey?

Thank you for this suggestion. We have explained better the application of this technique, and how we encourage respondents to pass along the link. The social networks used are Facebook and Instagram. For the first one, the link was shared on two groups (with similar number of members) respondents were invited to re-share the link one time before the compilation; on the second one, the questionnaire was spread out by 2 influencers with a similar number of followers (people which earn money using social networks), and they invited their followers to re-share the link one time.

The number of respondents is small (taking into account that this is not a specific samples with specific inclusion criteria but rather random). Power of samples size must be presented and sample size must be justified as well as it must be indicated in limitation section as a potential bias.

Thank you for this comment. In order to avoid bias due to the small sample we have included the robust estimation of standard error, and we have indicated it as a potential bias on discussion section. Furthermore, we provide in the text references which justify the sample size.

The question form applied in the survey should be presented.

Thank you for this suggestion. We have provided the question form applied in the Material and Methods section, both in terms of formulation and form. Due to the nature of the analysis, it is a close question (yes or no).

Taking into account the fact, that authors tried to analyze not only consumer behaviors but habits, the proper tool should be presented.

Thank you for this comment. We have analysed the habits related to the consumption of Extra Virgin Olive Oil. Therefore, we asked to interviewees if they are habitual consumers of Extra Virgin Olive Oil, with which frequency they buy it from the producers, and if they consume the oil which they produce themselves. Furthermore, we asked to consumers their purchase frequency of Extra Virgin Olive Oil labelled by big brands and by private labels.

The p<0.10 – as a not significant (but tend to be significant) should be omitted due to low samples size (and resultant high risk of bias)

Thank you for this comment. We omitted the p<0.10 as not significant (but tend to be significant).

Results should be presented in clearly in paragraphs (e.g. as a summarizing the meaning of the result – not only as a repetition of the results presented in tables). Some words should be corrected (e.g., consumer think – We do not know what they think we just could say what they declare, therefore authors should use “consumer declared think” )

Thank you for this comment. We have summarised the results at the end of the last paragraph, and we have corrected them in accordance with the modifications done to the methodology.

The discussion section should be broaden. Authors should relate the findings to those of similar studies and point out the differences and similarities between the studies. Authors should add the appropriate references in this section. The limitation section should be expanded (for the sample size, data gathering, etc.)

Thank you for this suggestion. The discussion section was been broaden following your suggestion, and we implement the appropriate references to the section. The limitations due to the sample size were added.

This section must be improved. Authors should present the conclusions of the study not the other information.

Thank you for this comment, we have improved this section. We have presented the main conclusions, both political and managerial, of the study. We also have added a part regarding the technological innovation of Communication and Information Systems following the suggestion of the Editor.

The manuscript must be corrected according to the authors guideline!

Done

Reviewer 2 Report

I read the work, it is really interesting; is clear and easy to reading. It also follows a correct methodological approach.

The authors, with a little effort can make an excellent paper.

Entering into the merits of things, it is necessary:

- The "Abstract" is well written and well synthetize the paper's arguments;

- "Introduction" makes the research question quite clear;

- “Literature review” is quite complete;

- “Materials and methods” is OK;

- “Results” and “Discussion” draw the trajectories of job development.;

- “Conclusions” can be improved and connected with the reference literature

In addition, the literature needs to be improved with more recent papers. The scientific debate on these issues over the last years has been very intense with works of high scientific importance. The authors could benefit from the use of Scopus and WoS databases or:

Ballco, P., Gracia, A., Do market prices correspond with consumer demands? Combining market valuation and consumer utility for extra virgin olive oil quality attributes in a traditional producing country, 2020, Journal of Retailing and Consumer Services, 53,101999; Mazzocchi, A., Leone, L., Agostoni, C., Pali-Schöll, I., The secrets of the mediterranean diet. Does [only] olive oil matter?, 2019, Nutrients, 11(12),2941; Giannoccaro, G., Carlucci, D., Sardaro, R., Roselli, L., De Gennaro, B.C., Assessing consumer preferences for organic vs eco-labelled olive oils, 2019, Organic Agriculture, 9(4), pp. 483-494; Bimbo, F., Bonanno, A., Viscecchia, R., An empirical framework to study food labelling fraud: an application to the Italian extra-virgin olive oil market, 2019, Australian Journal of Agricultural and Resource Economics, 63(4), pp. 701-72; Panzone, L., Di Vita, G., Borla, S., D’Amico, M., When Consumers and Products Come From the Same Place: Preferences and WTP for Geographical Indication Differ Across Regional Identity Groups, 2016, Journal of International Food and Agribusiness Marketing, 28(3), pp. 286-313;

Author Response

Reviewer 2

“Conclusions” can be improved and connected with the reference literature

Thank you for this comment. We have improved the conclusions and added the main literature references to support the statements.

In addition, the literature needs to be improved with more recent papers. The scientific debate on these issues over the last years has been very intense with works of high scientific importance. The authors could benefit from the use of Scopus and WoS databases or: Ballco, P., Gracia, A., Do market prices correspond with consumer demands? Combining market valuation and consumer utility for extra virgin olive oil quality attributes in a traditional producing country, 2020, Journal of Retailing and Consumer Services, 53,101999; Mazzocchi, A., Leone, L., Agostoni, C., Pali-Schöll, I., The secrets of the mediterranean diet. Does [only] olive oil matter?, 2019, Nutrients, 11(12),2941; Giannoccaro, G., Carlucci, D., Sardaro, R., Roselli, L., De Gennaro, B.C., Assessing consumer preferences for organic vs eco-labelled olive oils, 2019, Organic Agriculture, 9(4), pp. 483-494; Bimbo, F., Bonanno, A., Viscecchia, R., An empirical framework to study food labelling fraud: an application to the Italian extra-virgin olive oil market, 2019, Australian Journal of Agricultural and Resource Economics, 63(4), pp. 701-72; Panzone, L., Di Vita, G., Borla, S., D’Amico, M., When Consumers and Products Come From the Same Place: Preferences and WTP for Geographical Indication Differ Across Regional Identity Groups, 2016, Journal of International Food and Agribusiness Marketing, 28(3), pp. 286-313; 

Thank you for this comment. We have added the references proposed and, following the suggestion of the Editor, we have added a section in the literature review about the new Communication and Information Technologies applied to the agri-food supply chain, in order to make the literature review and the whole paper more actual.

Round 2

Reviewer 1 Report

Authors have made a great effort to improve the manuscript. All my comments were incorporated. After review, I have one more comment - please remove first sentence in conclusion section as it is not associated with the science purpose „The implications of the current study are mainly political.”

Author Response

Authors have made a great effort to improve the manuscript. All my comments were incorporated. After review, I have one more comment - please remove first sentence in conclusion section as it is not associated with the science purpose „The implications of the current study are mainly political.”

Thank you for this comment. Following your suggestion, we have modified the sentence highlighted (in red in the text) now the sentence is: “The study may be useful to understand which factors related to consumer behaviour influence the perception of Extra Virgin Olive Oil sustainability. This knowledge can be used by policy makers to plan actions in support of sustainability and its enhancement for the success of local business”
